# Zooming into Gut Dysbiosis in Parkinson’s Disease: New Insights from Functional Mapping

**DOI:** 10.3390/ijms24119777

**Published:** 2023-06-05

**Authors:** Luigia Turco, Nicola Opallo, Elisabetta Buommino, Carmen De Caro, Claudio Pirozzi, Giuseppina Mattace Raso, Francesca Lembo, Lorena Coretti

**Affiliations:** 1Department of Pharmacy, School of Medicine, University of Naples Federico II, Via Domenico Montesano 49, 80131 Naples, Italy; luigia.turco@unicampania.it (L.T.); nicola.opallo@unina.it (N.O.); elisabetta.buommino@unina.it (E.B.); claudio.pirozzi@unina.it (C.P.); mattace@unina.it (G.M.R.); lorena.coretti@unina.it (L.C.); 2Department of Precision Medicine, University of Campania Luigi Vanvitelli, 80138 Naples, Italy; 3Department of Science of Health, School of Medicine, University Magna Graecia of Catanzaro, Viale Europa, 88100 Catanzaro, Italy; decaro@unicz.it; 4Task Force on Microbiome Studies, University of Naples Federico II, 80100 Naples, Italy

**Keywords:** gut microbiota, Parkinson’s disease, PICRUSt2, short-chain fatty acids, quorum sensing

## Abstract

Gut dysbiosis has been involved in the pathogenesis and progression of Parkinson’s disease (PD), but the mechanisms through which gut microbiota (GM) exerts its influences deserve further study. Recently, we proposed a two-hit mouse model of PD in which ceftriaxone (CFX)-induced dysbiosis amplifies the neurodegenerative phenotype generated by striatal 6-hydroxydopamine (6-OHDA) injection in mice. Low GM diversity and the depletion of key gut colonizers and butyrate producers were the main signatures of GM alteration in this model. Here, we used the phylogenetic investigation of communities by reconstruction of unobserved states (PICRUSt2) to unravel candidate pathways of cell-to-cell communication associated with dual-hit mice and potentially involved in PD progression. We focused our analysis on short-chain fatty acids (SCFAs) metabolism and quorum sensing (QS) signaling. Based on linear discriminant analysis, combined with the effect size results, we found increased functions linked to pyruvate utilization and a depletion of acetate and butyrate production in 6-OHDA+CFX mice. The specific arrangement of QS signaling as a possible result of the disrupted GM structure was also observed. With this exploratory study, we suggested a scenario in which SCFAs metabolism and QS signaling might represent the effectors of gut dysbiosis potentially involved in the designation of the functional outcomes that contribute to the exacerbation of the neurodegenerative phenotype in the dual-hit animal model of PD.

## 1. Introduction

In our previous work, to mimic the multifactorial etiology of Parkinson’s disease (PD) and reproduce both the key central and peripheral features of the disease, we combined the striatal neurotoxin challenge using a 6-hydroxydopamine (6-OHDA) injection and ceftriaxone (CFX)-induced dysbiosis in a new dual-hit mouse model [1]. The oral administration of high dose of CFX results in gut dysbiosis, with a quantitative and qualitative alteration of gut microbial communities, reduction of beneficial microbes, and overgrowth of potential pathogens [2,3]. In the proposed two-hit model of PD, the CFX-induced dysbiosis amplified PD progression, worsening the motor deficits induced by intrastriatal 6-OHDA injection in mice [1]. Motor impairment was accompanied by a significant increase in neuronal dopaminergic loss, enhanced systemic and colon inflammation, and deterioration of the colonic architecture. In this mouse model, we followed the dynamics of gut microbiota by 16S rRNA gene sequencing to infer the role of dysbiosis in the exacerbation of PD pathology and symptoms. We detected a reduced gut microbiota richness in the two-hit mouse model, with a core microbiota depleted of common gut colonizers with probiotic activity and butyrate producers, namely *Butyricicoccus pullicaecorum* and *Papillibacter cinnamivorans*. Moreover, the mice were also depleted of *Bifidobacterium breve* and *Alistipes finegoldii*, the main acetate producers, suggesting the lack of cross-feeding interactions with the acetate-dependent butyrate producers [1].

Here, we implemented our analysis by focusing on definite functional potentials of the microbiota structure identified in the developed dual-hit PD model, with a possible implication in PD pathogenesis.

Several approaches are available to estimate the functional and metabolic contents of microbiota. The phylogenetic investigation of communities by reconstruction of unobserved states (PICRUSt) and its updated version PICRUSt2 [4] are validated bioinformatics tools that have been widely used to infer gene families, such as those found in the Kyoto Encyclopedia of Genes and Genomes (KEGG) orthologs and the Enzyme Commission (EC) numbers based on 16S rRNA gene sequencing data. Therefore, we sought to use PICRUSt2 to expand our previous study and lay the groundwork for bacterial cross-interaction, focusing on functional capability related to short-chain fatty acids (SCFAs) metabolism and quorum sensing (QS) signaling.

Gut microbe cross-feeding allows for several fermentation strategies to catabolize pyruvate into the main SCFAs: acetate, propionate, and butyrate. Gut-derived SCFAs exert physiologically important effects on host metabolism, ranging from the maintenance of intestinal barrier integrity and mucus production to anti-inflammatory outcomes [5,6]. The optimal balance of SCFAs production requires a well-structured microbiota in which members act collectively in multicellular groups with complementary metabolisms. 

QS is a population-level communication mechanism that connects microbial members and regulates their interactions in various microbial systems. QS is mediated by the production, release, and detection of extracellular signaling molecules called autoinducers (AIs). The QS language in microbial systems, such as gut microbiota, plays a significant role in various diseases because of its involvement [7] in gene expression relevant to the behavior coordination of collective cells, including biofilm formation, secondary metabolite synthesis, and virulence factor production [8].

In the present study, we used PICRUSt2 as an in silico mapping approach to unravel candidate pathways of cell-to-cell communication associated with CFX-induced dysbiosis and potentially involved in PD onset and progression in 6-OHDA-treated mice.

## 2. Results and Discussion

Our previous study characterized the dysbiotic gut microbiota profiles of the 6-OHDA model of PD in mice pretreated for 5 days with CFX by comparing the following experimental groups 14 days after 6-OHDA treatment: sham-operated mice receiving vehicle (Sham); mice receiving CFX for five days and subsequently unilaterally injected with the vehicle (CFX); PD mice lesioned by 6-OHDA intrastriatal injection (6-OHDA); mice receiving CFX for five days and subsequently lesioned (6-OHDA+CFX) (Figure 1A). In the present study, we aimed to explore the profiles of annotated functions related to SCFAs metabolism and QS in the same experimental groups to gain insight into the functional capacity of the bacteria identified by targeted sequencing of the 16S rRNA gene.

### 2.1. Mapping of Functions Related to SCFAs Metabolism and Acetate and Butyrate Fecal Level Quantification

First, we reviewed the functions related to SCFAs synthesis in the gut microbiota of each mouse using the Kyoto Encyclopedia of Genes and Genomes (KEGG) orthologues (KOs). According to Zhang et al., 556 KOs participating in the bioprocess of SCFAs were considered SCFAs-related KOs [9]. Among the total of 4975 KOs annotated, 212 were mapped into SCFAs-related KOs in our samples (Figure 1B). Then, we compared the abundance levels of the 212 SCFAs-related KOs across Sham, 6-OHDA, CFX, and 6-OHDA+CFX mice to identify the KOs functions specifically enriched or depleted in the 6-OHDA+CFX group. Based on linear discriminant analysis (LDA), combined with the effect size (LEfSe) results, 15 SCFAs-related KOs showed differences between Sham and 6-OHDA+CFX treated mice, all involved in acetate and butyrate production (nominal *p* < 0.05 by Kruskal–Wallis test, nominal *p* < 0.05 by pairwise Wilcoxon test, and logarithmic LDA score of 2.0). In particular, the 6-OHDA+CFX group was principally characterized by pyruvate-related Kos, such as K05396 (D-cysteine desulfhydrase), K01631 (2-dehydro-3-deoxyphosphogalactonate aldolase), K01714 (4-hydroxy-tetrahydrodipicolinate synthase), K01571 (oxaloacetate decarboxylase-Na+ extruding), K00656 (formate C-acetyltransferase), and depleted of succinate- and acetate-related KOs, mainly enriched in the Sham control group (K00226 and K01438, respectively; Figure 1C). The identified profiles clearly depicted a diverse metabolic disposition in SCFAs production among groups. Indeed, by quantifying the fecal content of acetate and butyrate, we obtained results clearly indicating a nominally significant reduction of acetate and butyrate levels in 6-OHDA+CFX mice with respect to the Sham controls (Figure 1D). 

Collectively, the results showed a reduction of acetyl-CoA/acetate-related KOs and acetate and butyrate levels in 6-OHDA+CFX mice, along with increased KOs linked to pyruvate utilization. Pyruvate, once produced, is catabolized in succinate, acetyl-CoA, and lactate to produce energy and/or SCFAs [10]. The reduction of acetyl-CoA/acetate-related KOs, the main pathway for butyrate biosynthesis, could indicate a decoupling of pyruvate utilization from butyrate production not sufficiently compensating for the butyrate demand. These metabolic alterations are likely to be directly managed by gut microbiota. In our previous study, we observed a dysfunctional gut microbiota in 6-OHDA+CFX mice, with its core depleted of common gut colonizers involved in the acetate-dependent butyrate production (*B. pullicaecorum* and *P. cinnamivorans*) and the main acetate producers (*B. breve* and *A. finegoldii*) [1], suggesting the lack of cross-feeding interactions concerning butyrate metabolism. These metabolic alterations might represent a crucial point deserving a deep investigation for clarifying the involvement of gut microbiota in PD progression. A growing body of evidence supports the idea that mitochondrial dysfunction, due to the alteration of pyruvate metabolism, might represent a key feature of PD [11,12,13,14]. Interestingly, abnormal pyruvate metabolism has been observed in PD patients [15,16], and increased serum pyruvate levels have been proposed as a PD biomarker, reflecting central dysfunction in both human patients and several animal models of PD, including 6-OHDA mice [17].

### 2.2. Mapping of Functions Related to QS

We aimed to study the distribution of predicted functions related to QS. Of the 4975 KOs annotated in our dataset, 84 were associated with the functional gene categories of the QS pathway, including autoinducer synthases and relevant QS receptors (https://www.genome.jp/entry/ko02024, accessed on 1 December 2022). LEfSe analysis showed that a total of 29 KEGGs discriminate between the 6-OHDA+CFX and Sham groups, of which 18 were increased and 11 decreased in the double-hit mice (Figure 2). Excluding 7/29 KEGGs that were specifically due to 6-OHDA or CFX treatment, the QS-related functions enriched in the 6-OHDA+CFX group were involved in the receptor recognition and intracellular signal transduction of AIs: K01626 (3-deoxy-7-phosphoheptulonate synthase) involved in phenazine biosynthesis; the GABA sensing proteins K01995-96 and K02053-55; the bacterial dipeptide permeases (Dpp) K02031-34; and the bacterial peptide permease (Opp) K10823, K15582-83. The KOs reduced in 6-OHDA+CFX were related to the first steps of QS, which include the bacterial production of AIs and their release, either actively or passively, into the surrounding environment: K03070, 73, 75 and 76, and K03210 (preprotein translocases); K03106 and K03110 (signal recognition particle subunit SRP54 and fused signal recognition particle receptor, respectively); K03210 (preprotein translocase subunit YajC); K03217 (YidC/Oxa1 family membrane protein insertase); and K12292 (ComA–competence factor transporting protein). Collectively, these data suggest an imbalance between the production and recognition of AIs in 6OHDA+CFX mice. Based on these exploratory results mapping bacterial functions, 6OHDA+CFX microbial communities showed reduced KOs related to AIs production and release mirroring the inadequately diverse gut microbiota harbored by these mice (Appendix A). This imbalance could impact community structure, complex communications, and in turn, microbiota functionality. As the microbiota richness drops, important AIs producers might decrease, inducing a compensatory increase in bacterial function associated with AIs receptors in the attempt to finalize environmental adaptation.

## 3. Materials and Methods

### 3.1. Establishment of Dual-Hit Mouse Model 

The overall experimental procedures to recapitulate and determine the dual-hit murine model of Parkinson’s disease are previously described in [1]. Briefly, ten-week-old male Swiss CD1 mice (20–25 g) were purchased from Harlan (Udine, Italy). They were housed in cages in a room kept at 22 ± 1 °C on a 12/12 h light/dark cycle. The animals were acclimated to their environment for 1 week and had ad libitum access to tap water and standard rodent chow. All animal experiments were conducted under license number no. 118/2019-PR, granted by the Italian Ministry for Health. All procedures involving the animals were carried out following the institutional guidelines and complied with the Italian D.L. (No. 26 of 4 March 2014) of the Italian Ministry for Health and associated guidelines from the European Communities Council (86/609/ECC and 2010/63/UE). The experimental procedure to induce PD was based on the unilateral intrastriatal injection of 6-OHDA (4 µg/2 µL) in mice. The mice were anesthetized (100 mg/kg ketamine plus 5 mg/kg xylazine intraperitoneally, i.p.) and placed in a stereotaxic apparatus (David Kopf Instruments, Tujunga, CA, USA) using a mouse adaptor and lateral ear bars. The head skin was cut longitudinally, and mice were injected unilaterally in the right striatum with 2 µL of 6-OHDA (2 µg/µL, Sigma Aldrich, Milan, Italy) solution using an injector connected to a 2 µL Hamilton syringe with polyethylene tubing. The 6-OHDA was diluted in 0.9% sterile saline with 0.2% ascorbic acid, filtered inside a laminar flow hood to avoid contamination, and protected from light throughout the procedure. The sham mice were injected with the same volume of 0.9% sterile saline and 0.2% ascorbic acid. The following stereotaxic coordinates were used: +0.5 mm anterior to bregma, −2.0 mm lateral to midline, and 3.5 mm ventral from the skull surface [18]. Once the 2 µL were injected, the syringe was kept in place for 5 min before being very slowly retracted from the brain, in no less than 5 min. To induce gut microbiota dysbiosis, the mice were treated with CFX (8 g/kg, per os) once daily, as previously reported [2], and five days later, the mice were unilaterally injected with 6-OHDA in the right striatum [19]. At 14 days, fecal samples were collected after defecation and stored at −20 °C prior to analysis. 

### 3.2. Genomic DNA Extraction, Illumina MiSeq Sequencing of Barcoded 16S rRNA Gene Amplicons, and Sequence Data Processing

Fecal samples for microbiota analysis were collected from five mice for each group and were analyzed using the Quantitative Insights Into Microbial Ecology (QIIME2, version 2021.4) [20], as reported in the experimental design previously published [1]. FASTQ reads were filtered, dereplicated, denoised, merged, and assessed for chimeras to produce amplicon sequence variants (ASVs) using the DADA2 pipeline [21]. After processing, rare ASVs, present in at least two samples, were filtered out, resulting in 384,875 high-quality reads. Data were rarefied to the minimum library size of 3620 reads/sample, a sequencing depth considered adequate, as all curves reached ASVs detection saturation. Phylogenetic tree generation was performed and ASVs have been classified at different taxonomic levels using the Greengenes reference database at a confidence threshold of 99% [22]. 

### 3.3. In Silico Predicted Metagenomic Analysis Based on Microbiota Sequencing Data

The phylogenetic investigation of communities by reconstruction of unobserved states (PICRUSt2), based on the normalized ASVs table, was applied to investigate the functional metagenomic profile in each sample [4]. The PICRUSt2 pathway tool was used via the default pipeline using the “picrust2_pipeline.py” script (https://github.com/picrust/picrust2/wiki/PICRUSt2-Tutorial-%28v2.3.0-beta%29, accessed on 1 December 2022). PICRUSt accuracy across the samples was measured using the weighted nearest sequenced taxon index (NSTI). The NSTI score is calculated as follows: “For every OTU in a sample, the sum of branch lengths between that OTU in the Greengenes tree to the nearest tip in the tree with a sequenced genome is weighted by the relative abundance of that OTU. All OTU scores are then summed up to give a single NSTI value per microbial community sample” [23]. NSTI scores range from 0 to 1 and reflect the availability of reference genomes that are closely related to the most abundant microbe in each sample. The mouse fecal samples had a mean NTSI value of 0.148, which is in line with previous mammalian-associated microbiota samples showing a mean NSTI value of 0.14 ± 0.06 s.d. [23]. Based on our previously published results on the dual-hit model of PD, in which we found an alteration in SCFAs producers and utilizers in an altered network of microbial community interaction, we focused our analysis on the Kyoto Encyclopedia of Genes and Genomes ortholog abundances related to the main SCFAs and quorum-sensing (QS) mechanisms. Differences among groups were analyzed and visualized using the linear discriminant analysis (LDA) effect size (LEfSe) method [24] (LEfSe; nominal *p* < 0.05 by the Kruskal–Wallis test, nominal *p* < 0.05 by the pairwise Wilcoxon test, and logarithmic LDA score of 2.0). 

### 3.4. Fecal SCFAs Extraction

Briefly, 500 mg of fecal sample was mixed with 500 μL of sterile water and homogenized for 3 min, then centrifuged for 30 min at 12,000× *g* at room temperature (RT). The supernatants (solubilized feces) were filtered and transferred into a new tube, where 20 μL of 85% (*w*/*v*) phosphoric acid (H_3_PO_4_) was added and mixed for 5 min. For SCFAs extraction, anhydrous diethyl ether (DE) was added to the acidified fecal homogenate samples (1:1, *v*/*v*), vortexed, and centrifuged for 30 min at 12,000× *g* at RT. The DE layer (containing SCFAs) was transferred into a new glass tube, where sodium sulfate anhydrous was added in order to remove the residual water. Finally, the organic phase was placed in a new glass tube for gas chromatography–mass spectrometry (GC-MS) analysis. For acetic and butyric acids, a standard curve (10–200 μg/mL) was generated at the beginning of the run. A blank solvent (DE) was injected between every sample to ensure that there were no memory effects.

### 3.5. Gas Chromatography–Mass Spectrometry (GC/MS) Analysis

The GC column was an Agilent 122-7032ui (DB-WAX-U, Agilent Technologies, Santa Clara, CA, USA) column of 30 m, with an internal diameter of 0.25 mm, and a film thickness of 0.25 μm. The GC was programmed to achieve the following run parameters: the initial column temperature was set at 90 °C, held for 2 min, and then increased to 100 °C at a rate of 2 °C/min, held for 10 min, and finally increased by 5 °C/min up to a final temperature of 110 °C for a total run time of 21 min, with a gas flow of 70 mL min.

### 3.6. Statistical Analysis

Data are presented as mean ± SEM. Differences in acetate and butyrate levels have been evaluated by using analysis of variance (ANOVA), followed by Tukey’s post hoc test. In this exploratory study, findings with *p* < 0.05 are reported as nominally significant results (without correction for multiple testings).

## 4. Conclusions

In this study, we pointed out the role of SCFAs and QS as possible mediators and modulators of the microbiota–host interaction in the two-hit (dysbiosis and striatal neurotoxin challenge) animal model of PD. Furthermore, a possible two-way cooperative action can be envisioned, based on the obtained results. 

Besides their well-known effects in maintaining intestinal homeostasis and contributing to gut immunity and gut–brain signaling, a role for SCFAs in affecting QS has been proposed. Acetic acid and other SCFAs influence biofilm formation, regulate the expression of virulence genes, inhibit bacterial growth, and bacteriocin production through QS modulation [25,26,27,28]. On the other hand, QS coordinates the cooperative expression of pyruvate metabolism genes to maintain a sustainable environment [29] and helps gut homeostasis with an influence on distal and central functions. Specifically, QS signals at low cell density can hinder the production of energy or SCFAs via the acetate synthesis pathway and favor the switch to pyruvate flux in order to prevent an unsustainable ecosystem as a result of metabolic activities (the production of neutral molecules versus the acidification of the extracellular environment) [29]. Accordingly, in 6-OHDA+CFX mice, we found a depletion of the cooperative production of acetate and butyrate, along with increased KOs linked to pyruvate utilization. The metabolic profile of the gut microbiota observed in the two-hit animal model of PD could be driven by QS signaling constraining the environmental bacterial adaptation in the context of destabilized community structure and diversity, lacking key species that sustain and preserve the community setting and the symbiosis with the host.

Thus, SCFAs and QS signaling might cooperatively represent the effectors of gut dysbiosis, possibly able to cause functional outcomes that contribute to the exacerbation and progression of the neurodegenerative phenotype in the dual-hit animal model of PD.

Our previous study has increased the knowledge of the role of gut dysbiosis in the progression and worsening of PD pathophysiology, but the effectors and mechanisms through which microbiota exerts its influences call for further studies. We provide a deeper picture of SCFAs and QS functional capability in the PD mouse model that requires further experimental validation. This work represents an explorative study to be confirmed by more extensive analyses, taking into account a larger sample size and broader metagenome functional predictions, in order to generate more accurate results to support the relationships found in our work using a rigorous methodological approach. In this view, the study of bacterial cross-interaction may enable new hypotheses for the experimental exploration of microbiota contribution to PD etiology and progression.

## Figures and Tables

**Figure 1 ijms-24-09777-f001:**
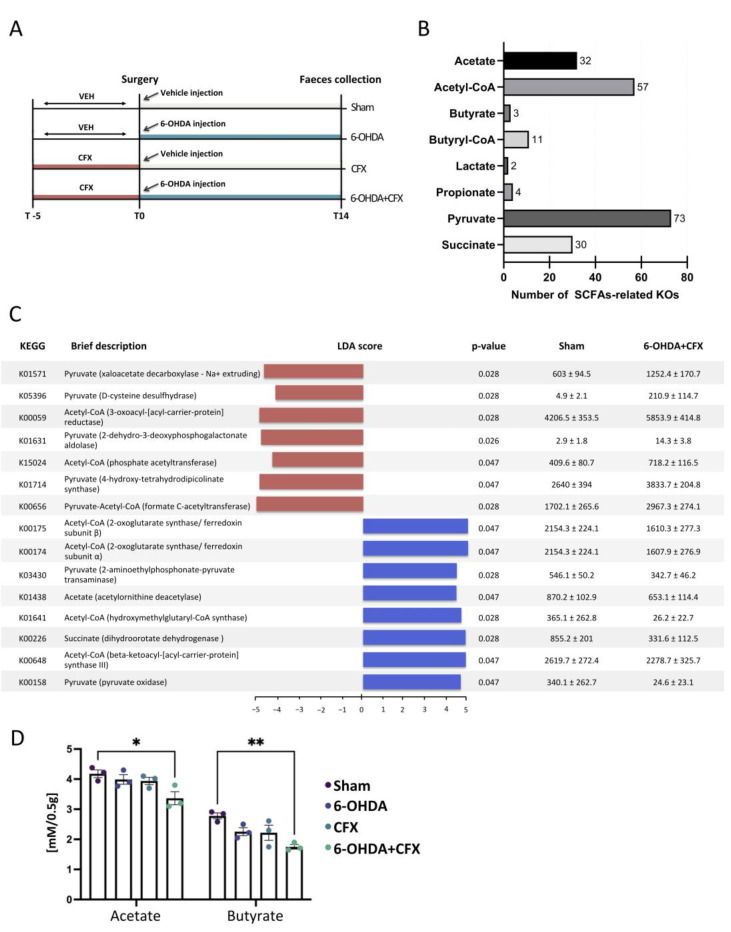
(**A**) Gut microbiota dysbiosis was induced through ceftriaxone (CFX, 8 g/kg/die, per os) administration, and five days later, the mice were unilaterally injected with 6-OHDA (4 µg/2 µL) in the right striatum. Sham mice were injected with the same volume of 0.9% sterile saline and 0.2% ascorbic acid at the same time point. After 14 days, feces were collected to perform gut microbiota analysis, mapping of SCFA- and QS-related KOs, and quantification of fecal SCFAs content. (**B**) The bar chart shows the distribution of a total of 212 SCFAs-related KOs, mapped into each experimental group, among the main SCFAs-related compounds. (**C**) Fifteen SCFAs-related KOs were found to be differentially abundant between Sham and 6-OHDA+CFX treated mice based on the LEfSe algorithm (nominal *p* < 0.05 by Kruskal–Wallis test, nominal *p* < 0.05 by pairwise Wilcoxon test, and logarithmic LDA score of 2.0). For each SCFAs-related KO identified, KEGG orthology, brief description, LDA scores, *p*-value, and relative abundance (mean ± STD err; *n* = 5 mice/group) in the Sham and 6-OHDA+CFX mice are reported. (**D**) Acetate and butyrate levels in fecal samples. Results are shown as mean ± STD err (*n* = 3 mice/group). Differences have been evaluated by ANOVA followed by Tukey’s post hoc test for multiple comparisons, * *p* < 0.05 and ** *p* < 0.01.

**Figure 2 ijms-24-09777-f002:**
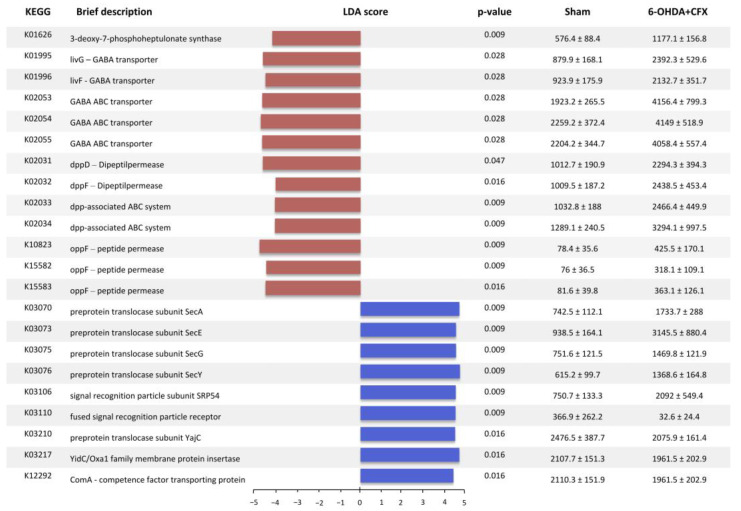
Differences among groups were analyzed and visualized using the LEfSe method (nominal *p* < 0.05 for both the Kruskal–Wallis and pairwise Wilcoxon tests, with an LDA score cutoff value (log10) above 2.0; *n* = 5 mice/group). For each key QS-related function, the KEGG orthology, a brief description, LDA scores, *p*-value, and relative abundance (mean ± STD err) in the Sham and double-hit mice are reported.

## Data Availability

Not applicable.

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
