# Peer review of "Zooming into Gut Dysbiosis in Parkinson’s Disease: New Insights from Functional Mapping"

_ijms, 2023, doi:10.3390/ijms24119777_

Round 1
Reviewer 1 Report
Dear All,
please find attached the Comments and Suggestions.
Kind Regards

Reviewer 2 Report
The study by Turco et al map 16S microbiome data of different experimental groups of mice (Parkinson/dysbiosis model: Sham, 6-OHDA, CFX, 6-OHDA+CFX) onto KEGG orthogonals and test group differences in LDA scores of these orthogonals.
The study has major issues related to statistical reporting and possibly appropriateness of methods that need clarification before publication:
Major issues:
1) How many mice (per group) are the results actually based on? This is not mentioned in the Methods. Sample size and data distribution are important to evaluate the appropriateness of the statistical/analytics methods used. Only Figure 1D caption says: “Results are shown as mean ± STD err (n = 3).” Do groups only contain 3 mice each?
2) The statistical/analytics process is only vaguely described. It should be noted in the abstract and methods that analyses are exploratory and did not account for multiple testing only accounted for the group comparisons and not the number of selected QS/SCFA-mechanism KOs. In Figure 2 the authors state “Significantly changed QS-related KOs between 6-OHDA+CFX and Sham groups”. The differences are not significantly different after accounting for multiple testing of >200 different KOs. Accordingly the interpretation should be more cautious.
“Differences among groups were analyzed and visualized using the linear discriminant analysis (LDA) effect size (LEfSe) method [20] (LEfSe; p < 0.05 by Kruskal–Wallis test, p < 0.05 by pairwise Wilcoxon test and logarithmic LDA score of 2.0).”
Minor issues:
1) To pick up the reader the beginning of the Introduction should be improved.
„…ceftriaxone (CFX)-induced dysbiosis amplified PD progression, worsening motor deficits induced by intrastriatal 6-hydroxydopamine (6-OHDA) injection in mice [1].” Please explain, how
ceftriaxone (CFX) induces dysbiosis, and which aspects mean dysbiosis
2) „PICRUSt2 as a predictive in silico approach“ please omit the word “predictive” as merely a mapping of 16S data on KEGG KO is performed. Please rephrase the sentence and the title, eg by replacing prediction with mapping
3) Please explain abbreviations when first occurring in the text, e.g. LEfSe and LDA
Only minor issues.
Round 2
Reviewer 2 Report
The authors addressed the issues raised.
Minor issue: Insert "nominal" before "significant" in p.3 L. 122.
Also, please rephrase p.7 L. 285 as " In this exploratory study, findings with p<0.05 are reported as nominally significant results (without correction for multiple testing)"
adequate
Author Response
We thank the Reviewer for the useful comments. We modified the manuscript accordingly to the suggestions.